# The Effects of Varying Glenohumeral Joint Angle on Acute Volume Load, Muscle Activation, Swelling, and Echo-Intensity on the Biceps Brachii in Resistance-Trained Individuals

**DOI:** 10.3390/sports7090204

**Published:** 2019-09-04

**Authors:** Christopher Barakat, Renato Barroso, Michael Alvarez, Jacob Rauch, Nicholas Miller, Anton Bou-Sliman, Eduardo O. De Souza

**Affiliations:** 1Department of Health Science and Human Performance, University of Tampa, Tampa, FL 33606, USA (C.B.) (M.A.) (J.R.) (N.M.) (A.B.-S.); 2School of Physical Education, University of Campinas, Campinas 13083-851, Brazil

**Keywords:** muscle length-tension relationship, bodybuilding, exercise selection, echo-intensity, muscle strain

## Abstract

There is a paucity of data on how manipulating joint angles during isolation exercises may impact overall session muscle activation and volume load in resistance-trained individuals. We investigated the acute effects of varying glenohumeral joint angle on the biceps brachii with a crossover repeated measure design with three different biceps curls. One session served as the positive control (CON), which subjects performed 9 sets of bicep curls with their shoulder in a neutral position. The experimental condition (VAR), varied the glenohumeral joint angle by performing 3 sets in shoulder extension (30°), 3 sets neutral (0°), and 3 sets in flexion (90°). Volume load and muscle activation (EMG) were recorded during the training sessions. Muscle swelling and strain were assessed via muscle thickness and echo-intensity responses at pre, post, 24 h, 48 h, and 72 h. There were no significant differences between conditions for most dependent variables. However, the overall session EMG amplitude was significantly higher (p = 0.0001) in VAR compared to CON condition (95%-CI: 8.4% to 23.3%). Our findings suggest that varying joint angles during resistance training (RT) may enhance total muscle activation without negatively affecting volume load within a training session in resistance-trained individuals.

## 1. Introduction

Physique athletes such as bodybuilders are primarily focused on maximizing muscle hypertrophy for their sport [1]. The literature implies that resistance training is the most effective method to increase muscle hypertrophy [2,3,4,5]. In addition, many RT variables (i.e. intensity, volume, exercise selection, joint angle, tempo) influence the acute responses [6,7]. Thus, coaches and athletes intentionally manipulate RT-variables within a session in order to optimize the session stimulus as well as training adaptations [1,8].

In this regard, recent evidence suggests that mechanical tension plays the greatest mechanistic role for muscle hypertrophy [3]. One objective variable that indirectly quantifies mechanical tension within a session is the volume load (sets × reps × load kg). While there is some controversy [9,10], it has been suggested that producing greater training volumes per session would optimize muscle mass accrual [5,11]. Additionally, the literature also suggests that other factors associated with metabolic stress (i.e. swelling, lactic acid accumulation) might impact skeletal muscle growth [2,3]. Furthermore, muscle activation is another key factor suggested to promote skeletal muscle adaptations [2,8,12,13,14]. Muscle activation is often analyzed by surface electromyography (EMG) and the data proposes the greater EMG amplitudes, the greater the motor unit recruitment [15]. Additionally, several acute variables may impact how muscle is activated during a RT session [15,16,17]. For example, relative intensity (% of one-repetition maximum (1RM)), total volume, rest intervals, and even joint angle affect muscle activation [15,16,18,19]. 

Interestingly, manipulating joint angle and RT at different muscle lengths has been shown to impact the acute responses associated with EMG and metabolic responses [7,19,20,21]. Altering joint angles influences the muscle length-tension relationship, thus affecting its ability to actively produce force [13,19,22,23]. Furthermore, overloading a muscle in its lengthened position will lead to an inefficient actin–myosin coupling (decreasing force output) while increasing the amount of strain and damage produced [7,21,23,24]. Nosaka et al. [7] reported significant increases in muscle damage and subjective soreness levels when training the elbow flexors at long muscle lengths compared to short muscle lengths in untrained individuals. While previous studies have investigated how the elbow joint angle and muscle length changes impact muscle activation [19,22,25,26], these studies had the shoulder joint in a fixed position (i.e., often flexed to 45 or 90 degrees using an isokinetic dynamometer), with minimal data on how glenohumeral angle would affect muscle activation of the biceps.

Only one study has investigated the effects of varying glenohumeral angles on muscle activation of the biceps. Oliveira et al. [18] reported differences in muscle activation of the biceps brachii in three RT exercises (e.g., incline dumbbell curl, standing dumbbell curl, dumbbell preacher curl). However, the use of different dumbbell exercises may have limited our understating on how varying glenohumeral joint angles would impact biceps brachii activation. For example, they reported the least amount of muscle activation when the elbow is fully flexed during the dumbbell preacher curl. This was likely due to the vertical force of the dumbbell and its inability to produce torque in that position of that particular exercise, not because of the glenohumeral joint angle. Therefore, the muscle activation differences reported would be explained by the variation in the resistance profile between those free-weight exercises rather than the variation in shoulder angle per se. While the aforementioned study provided insight on differences in muscle activation with different exercises, the effects of varying joint angles are not fully understood. 

More recently, Marcolin et al. [27] examined differences in muscle activation of the biceps brachii and brachioradialis while performing three forearm variants of curl (dumbbell, EZ bar, and straight bar) in resistance-trained individuals. They found differences in activation between all three exercises, with the EZ bar producing the greatest muscle activation on the biceps brachii even though glenohumeral joint angle was unchanged (0 degrees) between the three forearm variants. So, although differences in muscle activation are clearly multifactorial, it has not been investigated how varying joint angles throughout a training session would impact total muscle activation compared to an exercise utilizing a single joint angle. Yet, there is a paucity of data that examines how manipulating the joint angle through greater exercise variation affects session volume load, and metabolic stress factors in resistance-trained individuals. 

Therefore, the purpose of our study was to investigate how altering glenohumeral joint angle would affect session volume load, muscle activation, swelling and strain of the biceps brachii in resistance-trained individuals. We hypothesized that varying glenohumeral positions would enhance the acute training response by increasing total muscle activation and inducing more strain on the biceps. 

## 2. Materials and Methods

### 2.1. Experimental Design

This was a randomized crossover repeated measures design, which investigated the effects of altering glenohumeral joint angle with three different cable bicep curls (Free Motion, Logan, UT, USA) on session volume load, muscle activation, acute muscle swelling, and echo-intensity. One session served as a positive control (CON) in which all nine working sets of elbow flexion were performed with the shoulder in the neutral position (0°). In the experimental session (VAR), participants performed three sets of elbow flexion in three different glenohumeral joint positions (-30°, 0°, 90°), (Figure 1). The order in which the subjects performed each biceps curl variation was randomized. Each experimental session started with a 10 RM load that was predetermined and assessed during the familiarization sessions. Rest intervals between sets (60 s) and repetition tempo (2:2) were held constant amongst both conditions. Volume load (i.e., sets × reps × load (kg)) was assessed for each session to examine if altering glenohumeral joint angle would impact total work output. Muscle activation was measured through surface electromyography (EMG). Acute muscle thickness and echo-intensity were measured via ultrasound at rest/pre, immediately post, 24 h, 48 h, and 72 h after the training. Measurements were performed on the dominant arm and sessions were interspersed by one week and the subjects performed the remaining condition (e.g., CON or VAR). 

### 2.2. Subjects

Eleven subjects (5 males, 6 females) college students volunteered for this study (age: 21 ± 1.47 years, height: 166.8 ± 7.1 cm, body mass: 66.6 ± 10.4 kg, resistance training experience: 4.7 ± 1.91 years, 10 RM: 9.3 ± 3.1 kg). Inclusion criteria consisted of having at least 1 year of RT experience (defined as a minimum of three RT sessions per week) for males and females between the ages of 18–30. Subjects were excluded from participation if they were currently taking any medications, anti-inflammatory drugs, or had a history of drug abuse. All subjects reported no previous history of neck or upper extremity injury, and no surgical history. This study was approved by the university research ethics committee. All subjects read and willingly signed informed consent form by the Institutional Review Board. 

### 2.3. Familiarization

Prior to data collection, each subject’s dominant arm was identified. All subjects performed three familiarization sessions interspersed by 72 h prior to the commencement of the study. During familiarization sessions, subjects underwent 10-repetition maximum (10 RM) testing bilaterally on the cable bicep curl exercise in three different glenohumeral angles. The three-glenohumeral joint angles were shortened (flexion to 90°), neutral (0°), and lengthened (−30°). Each position was measured with a goniometer (Baseline® Evaluation Instruments, White Plains, NY, USA) by a certified athletic trainer and glenohumeral joint angle was maintained throughout the entire set without artificial stabilization. Subjects exercise execution was strictly enforced and visually monitored by the researchers throughout the training session to avoid possible cheating. Testing consisted of three separate 10-RM strength assessments, one for each glenohumeral angle as described above. 

### 2.4. Experimental Sessions

Subjects underwent two different training sessions interspersed by one week. Each session consisted of 9 working sets of cable biceps curls with one session serving as the positive control (CON) and one serving as the experimental session (VAR). During the CON session, subjects were instructed to maintain a neutral shoulder position (0 degrees) for the entire session. With the shoulder at 0 degree and the elbow at 90 degrees, subjects started the session with a 5-s maximum voluntary isometric contraction (MVIC) for elbow flexion and then rested for 60 seconds before performing one warm-up set of 10 repetitions with 50% of their 10 RM load which was determined during their familiarization sessions. This MVIC and submaximal set was performed as a means to normalize the EMG data. From there, subjects performed their first working set with their 10 RM. Repetition cadence was held constant throughout the session with a 2-s concentric and 2-s eccentric controlled by a metronome (EUMLab, Xanin Tech, Hangzhou, China). Subjects rested for 60-s in between sets. As fatigue set in and subjects could no longer complete 10-repetitions their load was reduced on the subsequent set to keep them within an 8–10 repetition range while performing each set at maximum intensity (e.g., 8–10 RM). During the VAR session, subjects started off each position with an MVIC and submaximal set (50% of their 10 RM). Thereafter, they performed three working sets in each angle at maximum intensity with the same repetition cadence and rest interval (60-s) as the positive control (CON). A certified strength and conditioning specialist and a certified athletic trainer were providing verbal encouragement and ensuring proper exercise execution throughout the duration of the study. 

### 2.5. Overall Session EMG

Muscle activation (EMG) was recorded using a 16-channel electromyography system (Trigno, Delsys, Boston, MA, USA) with an acquisition frequency of 2000 Hz and a hardware band-pass filter of 20–450 Hz. The skin area was shaved, abraded, and cleaned with an isopropyl alcohol pad to reduce skin impedance before electrode placement. One active bar wireless electrode (10 mm center to center, size: 27 × 37 × 13 mm, weight: 14 g, Trigno, Delsys, Boston, MA, USA) was placed on each subject’s dominant arm at the mid-belly mark (37.5%) of the biceps (Figure 2). The electrode was positioned parallel to the presumed orientation of the muscle fibers. The position of each electrode during the first session was marked on the skin with a henna tattoo. EMG signals were acquired in bipolar mode. In order to normalize the EMG data, a maximum voluntary isometric contraction (MVIC) was performed at each shoulder position followed by a submaximal set (50% 10 RM) before starting the experimental working sets in that position. We determined maximal muscle activation during the MVIC selecting a 500 ms window in which EMG values were maximal. In addition, as there is high between day variations in EMG, this procedure was performed on each session, therefore our EMG was expressed as the percentage of the maximal activation on each experimental session. The raw electromyography signals were digitally filtered (4th order Butterworth, band pass 20–500 Hz) and converted to root mean square (RMS). For the dynamic contractions, RMS was calculated for the entire set and normalized by the highest values obtained during the isometric contraction in 500 ms windows. Average EMG data was calculated for the entire session (i.e., overall session EMG).

### 2.6. Muscle Thickness

Ultrasonography (GE LOGIQ, General Electric Company, Fairfield, CT, USA) was used to assess muscle thickness (MT) of the elbow flexors of each subject’s dominant arm using a linear array probe (GE LOGIQ, General Electric Company, Fairfield, CT, USA) with a frequency of 8.0 MHz. To obtain b-mode images, subject laid supine in anatomical position, with their shoulder in external rotation and forearm supinated. The ultrasound probe was applied perpendicularly to the skin for measurement. A water-soluble gel (AQUASONIC® 100, Parker Laboratories, Inc., Fairfield, NJ, USA) was used on the transducer to aid acoustic coupling and remove the need for excess contact pressure on the skin. MT was defined as the distance between the interface of the muscle tissue and subcutaneous fat to the humerus. Two different areas were measured at 25% (distal) and 37.5% (mid-belly) of distance from the olecranon to the acromioclavicular joint. MT was assessed at rest/pre, immediately post, 24 h, 48 h, and 72 h post after exercise in order to assess changes in muscle swelling. To increase test-retest consistency, each site was marked with henna. To further ensure accuracy of the MT assessments, at least 3 images were obtained for each site. The median of the 3 assessments was used for statistical analysis. The coefficient of variation (CV) was determined prior to the start of the study using five different subjects with similar characteristics to those in the study. The CV for muscle thickness assessments was 1.3%. The same-blinded researchers performed sonography an all assessments. 

### 2.7. Echo Intensity

The echo intensity was measured using the same ultrasound device, probe, and standardizations already described. The echo intensity was determined by gray-scale analysis using the standard histogram function in Image-J (National Institute of Health, Laboratory for Optical and Computational Instrumentation, Madison, WI, USA, version 1.37). A region of interest (ROI) was chosen in each scan to include as much muscle as possible without any bone or fascia. The echo intensity in the region of interest was expressed in values between 0 and 256 (0: black; 256: white). Similar to the MT assessments, echo intensity analysis was performed at rest/pre, immediately post, 24 h, 48 h, and 72 h post session. For the echo intensity, the same-blinded experienced researcher analyzed all ultrasound images. Echo intensity of the biceps brachii was assessed to examine potential regional differences in fluid accumulation, strain, and potential muscle damage across conditions.

### 2.8. Statistical Analysis

Shapiro–Wilk testing confirmed that dependent variables were normally distributed. An analysis of variance (ANOVA) with repeated-measures was used to scrutinize the effects of varying glenohumeral joint angle on acute muscle thickness and echo intensity assuming condition (i.e., control and varying) and time (i.e., pre, post, 24 h, 48 h, and 72 h) as fixed factors. Whenever, a significant *F*-ratio was obtained, a post-hoc test with a Tukey´s adjustment was performed for multiple comparisons purposes. A paired *T-test* was used to compare the effects of conditions on volume load and overall session EMG. The significance level was previously set at p < 0.05. The 95% confidence intervals (95%-CI) were presented for the significant comparisons. The GraphPad Prism 8® was used to perform statistical analyses. 

## 3. Results

### 3.1. Volume Load and Overall Session EMG 

There were no significant differences in volume load between VAR and CON conditions (VAR: 596 kg ± 170 kg vs. CON 606 kg ± 175 kg, p = 0.59), Figure 3A. On the other hand, the overall session EMG amplitude was significantly higher (p = 0.0001) in VAR compared to CON condition (95%-CI: 8.4% to 23.3%), Figure 3B.

### 3.2. Muscle Swelling and Echo Intensity

For mid-belly muscle swelling, there was a main time effect (p = 0.0001) indicating that muscle swelling was greater at post compared to baseline (95%-CI: 0.35 to 0.56 cm). In addition, ES analysis revealed moderate acute effects for both VAR and CON conditions (ES: 0.54 and 0.43), respectively. The muscle swelling returned to baseline 24 h post similarly across conditions. For distal muscle swelling, there was a significant main time effect (p = 0.0001) indicating that muscle swelling was greater at post compared to baseline (95%-CI: 0.33 to 0.52 cm). In addition, ES analysis revealed moderate acute effects for both VAR and CON conditions (ES: 0.41 and 0.61), respectively. The muscle swelling returned to baseline 24 h post similarly across conditions. The individual responses for changes at the 25% (distal) and 37.5% (mid-belly) landmarks from pre to post, 24, 48, and 72 h workout are presented in Figure 4.

For mid-belly echo intensity (MBEI), there was a significant main time effect (p = 0.0001) indicating that EI was greater at post compared to baseline (95%-CI: 5.67 to 26.05 A.U). In addition, ES analysis revealed larger effects for both VAR and CON conditions (ES; 1.49 and 1.24), respectively. The MBEI returned to baseline 24 h post similarly across conditions. For distal echo-intensity (DEI), there was a significant main time effect (p = 0.0001) indicating that EI was greater at post compared to baseline (95%-CI: 4.69 to 22.29 A.U). However, ES analysis suggests that magnitude of effect was larger in the VAR when compared to CON condition (ES; 1.04 vs. 0.60), respectively. The EI returned to baseline 24 h post similarly across conditions. The individual responses for changes at the 25% (distal) and 37.5% (mid-belly) landmarks from pre to post, 24, 48, and 72 h workout are presented in Figure 5.

## 4. Discussion

The purpose of our study was to investigate how altering glenohumeral joint angle would affect session volume load, muscle activation, acute swelling, and echo-intensity of the biceps brachii in resistance-trained individuals. We partially confirmed our hypothesis as our main findings indicate that varying glenohumeral joint angles increased muscle activation without impacting total volume load compared to the control condition. Additionally, the magnitude of effect for echo-intensity at the 25% (distal) biceps suggests a greater response immediately post-workout in the VAR condition. However, both conditions responded similarly and returned to baseline levels 24 h post. Furthermore, there was a similar response between conditions regarding acute muscle swelling.

### 4.1. Volume Load and Overall Session EMG

Research has suggested a dose response relationship between training volume load and increases in muscle mass [10,11,28,29]. However, it is noteworthy to mention that there is no study comparing the effects of altering joint angles on session volume load. Therefore, it is difficult to compare our study to current literature. In this acute study, both conditions demonstrated similar volume load (e.g., VAR: 596 kg ± 170 kg, CON 606 kg ± 175 kg, p = 0.59) working at the same intensity. Although it is understood that altering the muscle lengths can impact its ability to produce force [13,23], our data suggest that trained individuals can maintain volume loads across greater joint angle variety with isolation movements (e.g. elbow flexion) within a training session. 

Interestingly, despite similar volume load, overall session EMG was different between conditions. Our results demonstrated that VAR condition produced greater total muscle activation compared to CON (e.g., 95%-CI: 8.4% to 23.3%, p = 0.0001). This data highlights potential differences between internal work demands (muscle activation) and external work (volume load). Although it is well established in the literature that differences in muscle length influences motor unit recruitment and muscle activity [30,31], the findings are contradictory [19,25,32,33]. Thus, making our findings quite difficult to reconcile. However, our data suggest there is an acute benefit to varying glenohumeral joint angle when performing biceps curls. Therefore, further research is required to determine whether or not these differences in acute muscle activation would enhance chronic adaptations (i.e., muscular hypertrophy and strength).

### 4.2. Muscle Swelling and Echo Intensity 

Acute muscle swelling and echo intensity were used as markers of metabolic stress and muscle strain [6]. Regarding acute muscle swelling we reported an average (~12.2%) which corroborates with previous research that demonstrated ~13.5–15.0% increase in muscle swelling in strength-trained individuals [16,34]. However, the acute muscle swelling returned to the baseline 24 h post the training session. Our findings partially agree with Nosaka et al. [7] findings regarding muscle swelling and damage. Utilizing both circumference measurements and muscle thickness (i.e., swelling) assessments, they reported a significant main time effect from pre to post in both regions (mid-belly and distally). However, they reported larger increases distally, whereas we observed similar changes at both regions between conditions. Contrary to our findings, their muscle swelling did not return to baseline 24 h post and remained elevated. They reported the greatest increase in swelling 72 h post exercise in the distal region for their lengthened condition. Our conflicting results are due to the differences in subject population. Their subjects were described as nonathletes and not involved in regular RT (i.e., untrained). Whereas we investigated resistance-trained individuals with an average of 4.7 years of RT experience. It is well understood that well-trained individuals have significantly greater recovery capabilities compared to untrained [35]. For example, Newton et al. [35] observed significantly greater recovery on makers of muscle damage in trained subjects compared to untrained undergoing RT of the elbow flexors. 

In regards to echo-intensity, while we found a main time effect for both conditions from pre/rest to immediately post workout at both sites of the biceps (MBEI, DEI), echo-intensity was back to baseline 24 h post. Distally (DEI) the effect size analyses suggested that VAR condition produced more strain compared to CON (ES; 1.04 vs. 0.60). Previously, Nosaka et al. [7] reported significantly greater echo-intensity and plasma creatine kinase in their lengthened condition. Perhaps, their data shines light as to why this present study VAR condition demonstrated a larger magnitude of effect for DEI, as this condition performed three working sets in a lengthened position for the long head of the biceps (−30 degrees glenohumeral extension) that CON did not. However, it is noteworthy to mention that while previous studies have demonstrated that training in a lengthened position produced greater muscle damage, these investigations were in untrained individuals [21,35,36]. Our findings only suggest that exercising in a lengthened position places more strain during exercise leading to increases in echo-intensity immediately after the training session. However, more research is warranted to determine if exercising in a lengthened position would translate to better adaptations in resistance-trained-individuals.

### 4.3. Limitations

Our study has several limitations that need to be addressed. 1) EMG data can be influenced by multiple factors (i.e., subcutaneous tissue, spatial filter transfer function, innervation zone (IZ), electrode placement, etc.) and must be taken with caution [37]. 2) In particular, dynamic contractions in fusiform muscles (i.e., biceps brachii) are very difficult to study as any observed changes may be due to either muscle activation or the geometry of the electrode-muscle system (i.e., muscle lengthening or shortening, fiber overlap) [38] Additionally, during the concentric contraction of the biceps, the IZ shifts upwards, moving underneath the electrode, thus impacting amplitude values. 3) Our electrode placement may not have been optimal [38] as our mid-belly (37.5%) anatomical landmark frame (ALF) was determined based on the distance from the acromion to the olecranon and not the distance between the acromion and distal insertion of the biceps brachii. Moreover, the optimal electrode site for the short head and the long head of the biceps brachii has been identified as 61% and 62% of the ALF, respectively, where ours was placed on the muscle belly at 62.5% of our ALF for consistency sake between multiple assessments (EMG, muscle swelling, and echo-intensity). 4) We did not assess perceptual measures (i.e. RPE) between conditions to monitor potential exertion differences. 5) We did not evaluate the subject’s perceived soreness throughout the recovery period. 6) We had a small sample size (n = 11). 7) Lastly, this was an acute study. Further research is needed to examine the chronic effects of varying joint angles and its effects on RT-induced adaptations.

## 5. Conclusions

In conclusion, our findings suggest that varying joint angles during RT may enhance total muscle activation and can potentially induce more strain without negatively affecting volume load within a training session in resistance-trained individuals. From a practical application standpoint, resistance-trained subjects or bodybuilders trying to maximize the training stimulus of each session should utilize multiple exercises that vary joint angles. This may lead to a greater internal stimulus (muscle activation) while performing the same amount of total work (volume). However, we cannot draw conclusions on how this may influence chronic adaptations, as this was an acute study. 

## Figures and Tables

**Figure 1 sports-07-00204-f001:**
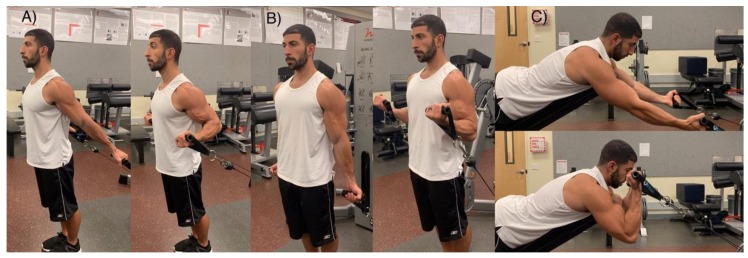
Glenohumeral joint angles—(**A**) -30° extension, lengthened; (**B**) 0°, neutral; (**C**) 90° flexion, shortened. Please confirm which form of degree you want to use: degree or the unit “°” and please use the same form for unification in the main text.

**Figure 2 sports-07-00204-f002:**
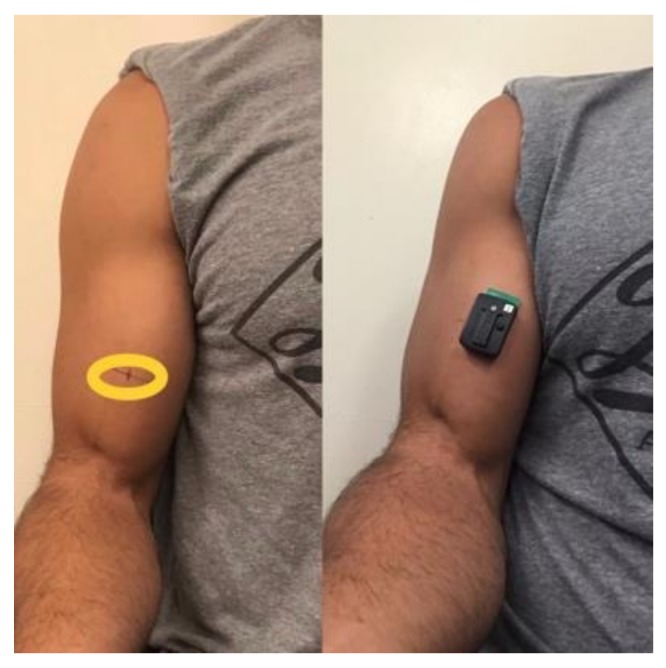
Mid-belly (37.5%) location site utilized for electrode placement, muscle thickness, and echo-intensity assessments.

**Figure 3 sports-07-00204-f003:**
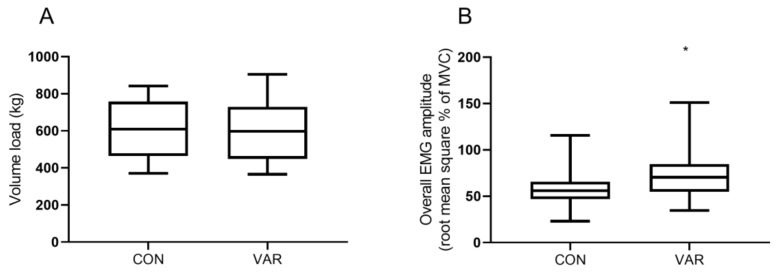
Total session volume load (**A**) and overall session surface electromyography (root mean square % of maximum isometric voluntary contraction) (**B**).

**Figure 4 sports-07-00204-f004:**
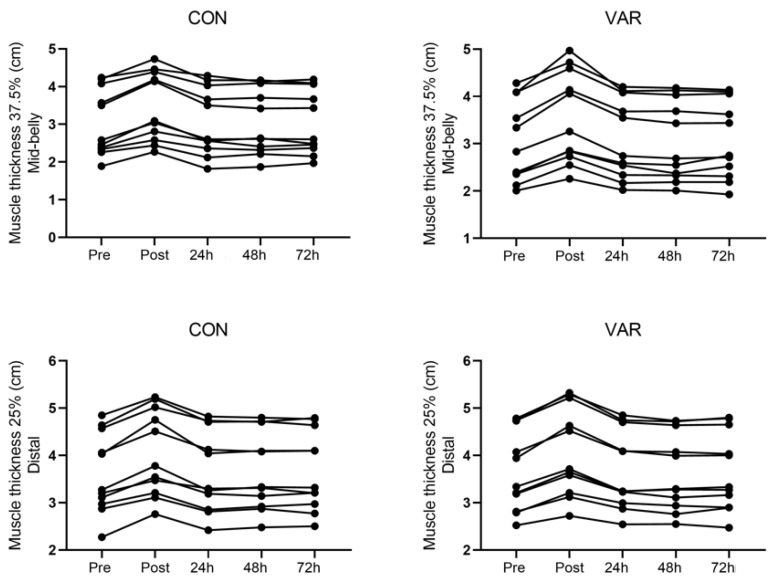
Muscle thickness (i.e., swelling) individual responses for changes at the 25% (distal) and 37.5% (mid-belly) landmarks from pre to post, 24, 48, and 72 h workout.

**Figure 5 sports-07-00204-f005:**
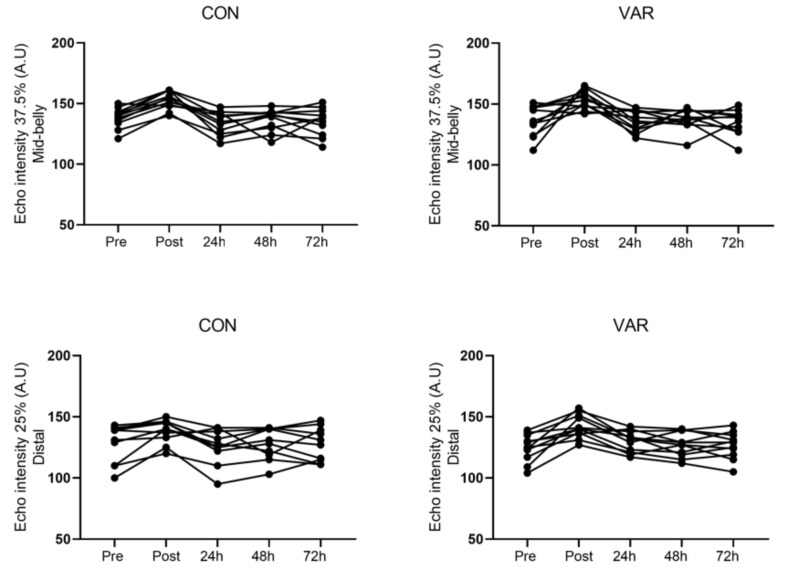
Echo-intensity individual responses for changes at the 25% (distal) and 37.5% (mid-belly) landmarks from pre to post, 24, 48, and 72 h workout.

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
