# Peer review of "The Effects of Varying Glenohumeral Joint Angle on Acute Volume Load, Muscle Activation, Swelling, and Echo-Intensity on the Biceps Brachii in Resistance-Trained Individuals"

_sports, 2019, doi:10.3390/sports7090204_

Round 1
Reviewer 1 Report
Manuscript ID: sports-576048
Title: Varying Glenohumeral Joint Angle Increases Muscle Activation in the Biceps Brachii without Impacting Volume Load in Resistance-Trained Individuals
This study investigated the issue of altering glenohumeral joint angle in relation to volume load, muscle activation, acute swelling and echo-intensity of the BB in RT individuals.
COMMENTS TO THE AUTHOR(S)
The most relevant and significant result of the study is the overall session EMG amplitude (RMS). Unfortunately, I noticed that the authors positioned the electrodes without paying attention to the innervation zone (IZ). Indeed, in surface EMG acquisitions this is a critical point, since the IZ may impacts dramatically on the amplitude of the signal and may invalidate the conclusions of the study. According to the Atlas of the innervation zones of Barbero, Rainoldi and Merletti (Springer 2012), the biceps IZ is located at 30% on a line between the cubital fossa and the acromion, which is very near to the distance indicated by the authors).
Moreover, during dynamic contractions, the IZ shifts, affecting even more the signal and its interpretation. Therefore, the IZ must be detected before the electrodes are placed on the muscle belly and that region must be avoided.
Other comments:
P3 L118 “glenohumeral joint was maintained throughout the entire set”. Please explain how this was done. By a electrical goniometer or using sensors? This seems a critical point and should be better described. Abbreviations should be define once and then used throughout the manuscript (e.g. electromyography, resistance training, repetition maximum, resistance-trained)
SPECIFIC COMMENTS
Introduction
P2 L61 recently the group of Paoli published promising results about EMG activity of the BB during three different exercises in RT participants. (Marcolin et al., 2018; DOI 10.7717/peerj.5165).
I suggest to modify the introduction by adding this reference.
Materials and Methods
P5 L173 The description of the electrodes positioning “37.5% of distance from olecranon to the acromioclavicular joint” does not match the recommendations suggested by the SENIAM guidelines (2000) or to Barbero et al (2012). Moreover, adding a picture of the electrode position may help the reading to identify the position used by the authors.
L194 please add city, state to the statistical software company
Results
P7 L225-6 please use the same order “VAR and CON” also in the parenthesis.
L226 Figure 3A does not exist. Probably it should be 4A
Figure 4 please increase the font size of the asterisk
Figure 4 caption: A and B description should be inverted according to the figure
Discussion
Section 4.2 should be expanded
Section 4.3
the small sample size may also affect the conclusions of the study and should be added as a limitation.
Reviewer 2 Report
Manuscript title
Varying Glenohumeral Joint Angle Increases Muscle Activation in the Biceps Brachii without Impacting Volume Load in Resistance-Trained Individuals
Major concerns
- The study compared between 3 different glenohumeral (GH) joint positions (i.e., at 30, 0 and 90 deg) (this is the Experimental or VAR trials) with the GH joint position at 0 deg (Control trial). This comparison between the experimental and control conditions were done ‘globally’, i.e., with the data of all the measures taken during the experimental trials, i.e., the data from the 3 GH joint angles were averaged out and then this mean data was then compared with the same measures taken in the Control trial. This reviewer feels that “combining all the data of the 3 GH joint angle in the experimental trial” is not really ideal or optimal/ appropriate because these data came are from different exercise positions. The Reviewer feel that the comparison should be between the 2 different glenohumeral (GH) joint positions (i.e., at 30 and 90 deg) with the GH joint position at 0 deg. This comparison, this reviewer feels, is a much fairer comparison and the results from this comparison has more real-world practical application. As an example, Line 278, it was stated “both conditions (i.e., between experimental and control trials; italics by reviewer) demonstrated similar volume load (e.g., VAR: 596 ± 170 kg, CON 606 ± 175 kg p = 0.59) working at the same intensity”. It would have been more ideal if the authors can say that when the GH joint angle was at 30 deg the amount of weight lifted was XXX kg, and when the GH is at 90 deg the amount of weight lifted was XXX kg, and lastly when the GH joint angle was at 0 deg the amount of weight lifted was XXX – these specific angles comparison information, this reviewer felt, is more beneficial and useful (rather than then global or total amount of the 3 experimental trials lifted that is currently being provided). With the specific GH joint angles comparison information, the individual who is thinking of doing biceps curls is now able to decide which of the 3 different GH joint angle allows one to lift more (and at the same time with the other measures that were taken such emg and echo-intensity, etc.; will henceforth allows the individual to make a good decision on what is the best exercise position to gain hypertrophy or strength in the biceps). The current manner in which the authors is doing the comparison of using the global data of the experimental (VAR) and CON will only apply if the individual want to perform the 3 different biceps curls exercises, i.e., at 3 different GH joint angels, during a single resistance exercise session – which is plausible but is uncommon for a sporting or recreational active individual (such a manner of doing different joint angles for the same muscle is however common in the sport of body-building where one exercises the same muscle group using various types of similar or related exerciser to try to stimulate and strain the same muscles over and over again). This reviewer Is incline to think his idea is better but please convince me otherwise.
- Statistics. Author use effects sizes to interpret results (line 262) and then use p-values for others. This lack of consistency made it appears as though the authors is picking and choosing what they want to see. To be consistent, authors should write down all information on both p-values and effects sizes for the comparison data and then provide some arguments and support why they believe one of the two stats is telling the “true”.
- The reviewer feels that the exercise volume data should be presented first in the Results section rather than any other data. This is because we need set the platform that the two sessions were similar in volume performed. Similarly, for the Discussion section, the training volume between the 2 conditions should be discussed first before any other data.
Minor concerns
- Line 40. You should define what is set, reps and weight here in the first instance before assuming every reader knows what this formula means. Also, in Line 93, you used different “sets x reps x load”.
- Line 50. Delete “training” and use “exercise”.
- Line 55. Delete “2000”. This is already written in the reference section. Check this error throughout the manuscript. Another one Line 257.
- Line 57. Add “muscle length” after the word “short”. This helps to make the sentence much clearer.
- Line 105. Need to be more specific to the subjects “resistance training experience”.
- Line 110. Were the subjects involved informed of the purpose or aim of the study? If they were, then there is possibility of a placebo effects during the trials (physical efforts put in by the subjects) on the outcome of the study and this should be highlighted as limitation of the study.
- All Figures. Poor resolution. Please improve.
- Figure 1. It would have been ideal, if the authors could also include a simple stick figure of the positions of the GH and elbow joints angle to accompany the photo – this will greatly help the reader to visual what is happening at these 3 different GH and elbow joint angles.
- Line 190 and 191. Use italic for F ratio and t-test.
- Line 201 and 206 is the same sentence repeated. Eliminate either one.
- Line 226. Shouldn’t it be Figure 4A? and Line 228 Figure 4B?
- Line 237-238. This sentence is awkwardly written – and I believe this is the most important finding of your study. Please rewrite this sentence again, carefully.
- Line 242-243. Has acute swelling been used as a marker of metabolic stress and/or muscle strain – a good reference is required here and more importantly; this point requires a study that has validated the relationship between acute muscle swelling and metabolic stress/muscle strain?
- Line 243. Is comparing with Nosaka study appropriate in this situation? – because did Nosaka used the same type of exercise (at the same GH and elbow joint angles). A possible major difference could be because in Nosaka study, the subjects performed more repetitions (or a higher volume of the exercise performed relative to the current study – rather because of differing training status as mentioned by the authors).
Line 292. The present study did not provide any practical applications of their findings (please refer to my major concerns comments).
Author Response
Reviewer 2
Varying Glenohumeral Joint Angle Increases Muscle Activation in the Biceps Brachii without Impacting Volume Load in Resistance-Trained Individuals
Major concerns
- The study compared between 3 different glenohumeral (GH) joint positions (i.e., at 30, 0 and 90 deg) (this is the Experimental or VAR trials) with the GH joint position at 0 deg (Control trial). This comparison between the experimental and control conditions were done ‘globally’, i.e., with the data of all the measures taken during the experimental trials, i.e., the data from the 3 GH joint angles were averaged out and then this mean data was then compared with the same measures taken in the Control trial. This reviewer feels that “combining all the data of the 3 GH joint angle in the experimental trial” is not really ideal or optimal/ appropriate because these data came are from different exercise positions. The Reviewer feel that the comparison should be between the 2 different glenohumeral (GH) joint positions (i.e., at 30 and 90 deg) with the GH joint position at 0 deg. This comparison, this reviewer feels, is a much fairer comparison and the results from this comparison has more real-world practical application. As an example, Line 278, it was stated “both conditions (i.e., between experimental and control trials; italics by reviewer) demonstrated similar volume load (e.g., VAR: 596 ± 170 kg, CON 606 ± 175 kg p = 0.59) working at the same intensity”. It would have been more ideal if the authors can say that when the GH joint angle was at 30 deg the amount of weight lifted was XXX kg, and when the GH is at 90 deg the amount of weight lifted was XXX kg, and lastly when the GH joint angle was at 0 deg the amount of weight lifted was XXX – these specific angles comparison information, this reviewer felt, is more beneficial and useful (rather than then global or total amount of the 3 experimental trials lifted that is currently being provided). With the specific GH joint angles comparison information, the individual who is thinking of doing biceps curls is now able to decide which of the 3 different GH joint angle allows one to lift more (and at the same time with the other measures that were taken such emg and echo-intensity, etc.; will henceforth allows the individual to make a good decision on what is the best exercise position to gain hypertrophy or strength in the biceps). The current manner in which the authors is doing the comparison of using the global data of the experimental (VAR) and CON will only apply if the individual want to perform the 3 different biceps curls exercises, i.e., at 3 different GH joint angels, during a single resistance exercise session – which is plausible but is uncommon for a sporting or recreational active individual (such a manner of doing different joint angles for the same muscle is however common in the sport of body-building where one exercises the same muscle group using various types of similar or related exerciser to try to stimulate and strain the same muscles over and over again). This reviewer Is incline to think his idea is better but please convince me otherwise.
Answer: We would like to thank the reviewer for his/her time and effort in reviewing our manuscript entitled "The Effects of Varying Glenohumeral Joint Angle on Acute Volume Load, Muscle Activation, Swelling and Echo-Intensity on the Biceps Brachii in Resistance-Trained Individuals” On this revised version, we attempted to address most of the points and edits brought up by the reviewer. The modifications throughout the manuscript in response to the comments listed below are highlighted in yellow.
In regard to the aforementioned major concern, we understand your thought process and concern here. However, our study design was intended to explore how varying joint angles in one session may have greater acute effect compared to training in with one joint angle. We also utilized a randomized design, so during the VAR condition, subjects started in different positions (-30,0, or 90) so comparing volume or muscle activation between the positions would be inaccurate due to difference in fatigue and time point of the session (i.e. sets 1-3 vs sets 7-9). For us to accurately assess the differences between each angle and compare them to one-another, we would have had to have two additional sessions where all 9 sets would be performed at one of the joint positions.
Although a lot of sport athletes may only utilize 1 exercise or 3 sets in one session for an isolation exercise like the biceps curl, as you mentioned a lot of bodybuilders use multiple exercises per session on the same muscle group. Hackett et al. reported that 74% of bodybuilders use 4-5 exercises per muscle group. Moreover,, some lifters (recreationally active or bodybuilders) may perform three different exercises with the same joint position (e.g. standing dumbbell curl, standing barbell curl, standing cable curl – shoulder neutral in all three exercises). Therefore, we wanted to investigate how varying joint angles within a training session would affect acute outcomes associated with long-term muscle growth compared to less variation within a training session..
We believe a lot more research is needed to better understand how varying joint angles may contribute to differences in the acute response, but we are trying to help fill the gap there.
Hackett, D. A., Johnson, N. A., & Chow, C. M. (2013). Training practices and ergogenic aids used by male bodybuilders. The Journal of Strength & Conditioning Research, 27(6), 1609-1617.
- Statistics. Author use effects sizes to interpret results (line 262) and then use p-values for others. This lack of consistency made it appears as though the authors is picking and choosing what they want to see. To be consistent, authors should write down all information on both p-values and effects sizes for the comparison data and then provide some arguments and support why they believe one of the two stats is telling the “true”.
Answer: Several reviews papers have suggested not only the use of null hypothesis testing to interpret the study’s findings. Additionally, it has been recommended to use uncertainty measurements (confidence intervals) and the measure of the magnitude (effect sizes) to further explore the data. It is noteworthy to mention, we have used p values for all our interpretations and we decided to call the attention for effects sizes only for one variable (i.e. echo-intensity). Yet, this was done in a very conservative manner. Therefore, we feel the statistical analysis used and our interpretations were well reflected the data and our findings.
- The reviewer feels that the exercise volume data should be presented first in the Results section rather than any other data. This is because we need set the platform that the two sessions were similar in volume performed. Similarly, for the Discussion section, the training volume between the 2 conditions should be discussed first before any other data.
Answer: Thank you very much for that suggestion. We have made those adjustments and kept the order consistent throughout (intro, methods, results, discussion). Our introduction mentioned volume and muscle activation first, so we should have presented our data as you suggested from the start.
Minor concerns
- Line 40. You should define what is set, reps and weight here in the first instance before assuming every reader knows what this formula means. Also, in Line 93, you used different “sets x reps x load”.
Answer: Thank you for this observation. It was changed accordingly and made consistent throughout (load [kg]).
- Line 50. Delete “training” and use “exercise”.
Answer: We altered this to resistance training (RT) due to preference.
- Line 55. Delete “2000”. This is already written in the reference section. Check this error throughout the manuscript. Another one Line 257.
Answer: It was changed accordingly and kept consistent throughout.
- Line 57. Add “muscle length” after the word “short”. This helps to make the sentence much clearer.
Answer: It was changed accordingly.
- Line 105. Need to be more specific to the subjects “resistance training experience”.
Answer: We added more information regarding our inclusion criteria and the resistance training experience of our subjects.
- Line 110. Were the subjects involved informed of the purpose or aim of the study? If they were, then there is possibility of a placebo effects during the trials (physical efforts put in by the subjects) on the outcome of the study and this should be highlighted as limitation of the study.
- All Figures. Poor resolution. Please improve.
Answer: Photo Figures have been changed for improved resolution.
- Figure 1. It would have been ideal, if the authors could also include a simple stick figure of the positions of the GH and elbow joints angle to accompany the photo – this will greatly help the reader to visual what is happening at these 3 different GH and elbow joint angles.
Answer: With the improved resolution of the photos, we feel this best represents what was performed as we provided not just the shoulder position, but the beginning and end range of each repetition.
- Line 190 and 191. Use italic for F ratio and t-test.
Answer: It was changed accordingly.
- Line 201 and 206 is the same sentence repeated. Eliminate either one.
- Line 226. Shouldn’t it be Figure 4A? and Line 228 Figure 4B?
Answer: Thanks for catching that. It was changed accordingly and is now 2A and 2B as we revised the order of presenting our data, starting with volume, per your suggestion.
- Line 237-238. This sentence is awkwardly written – and I believe this is the most important finding of your study. Please rewrite this sentence again, carefully.
Answer: It was improved upon and changed accordingly.
- Line 242-243. Has acute swelling been used as a marker of metabolic stress and/or muscle strain – a good reference is required here and more importantly; this point requires a study that has validated the relationship between acute muscle swelling and metabolic stress/muscle strain?
Answer: We included a reference for this sentence. Yes, acute muscle swelling has been used as a marker for metabolic stress. De Freitas et al 2017 - Role of metabolic stress for enhancing muscle adaptations: Practical applications.
- Line 243. Is comparing with Nosaka study appropriate in this situation? – because did Nosaka used the same type of exercise (at the same GH and elbow joint angles). A possible major difference could be because in Nosaka study, the subjects performed more repetitions (or a higher volume of the exercise performed relative to the current study – rather because of differing training status as mentioned by the authors).
Answer: The exercise was slightly different as Nosaka had artificial stabilization of the GH joint. However, Nosaka et al. had subject perform 24 total reps where our subjects performed 72-90 total reps (24-30 reps in each position). Thus, our total volume was much higher.
Line 292. The present study did not provide any practical applications of their findings (please refer to my major concerns comments).
Answer: We have added a practical application component to our conclusion section.
Reviewer 3 Report
General comments:
The manuscript is well written and includes very practical outcomes for resistance training. However, there some points, where it looks like authors also compared EMG between different joint angles (e.g. beginning of discussion). I would write e.g.: Varying Glenohumeral Joint Angle sets (training session etc.) to be clear that there is no comparison between exercises. I also appreciate the straight forward approach of authors.
Line 20: please format the upper case for °.
Line 26-27: state more practical outcome
Line 34-35, 38-40: The exercise tempo is very important parameter for acute response as well.
https://www.ncbi.nlm.nih.gov/pubmed/29922395
Line 93: why you selected tempo 202?
Line 176-185: How did you avoided the electrode positioning differences between different angles?
https://www.ncbi.nlm.nih.gov/pubmed/16102976
Line 187: Please report the effect sizes since you have n = 11.
Line 226: the figure 4 is not reported.
Author Response
Reviewer 3
The manuscript is well written and includes very practical outcomes for resistance training. However, there some points, where it looks like authors also compared EMG between different joint angles (e.g. beginning of discussion). I would write e.g.: Varying Glenohumeral Joint Angle sets (training session etc.) to be clear that there is no comparison between exercises. I also appreciate the straight forward approach of authors.
Answer:We would like to thank the reviewer for his/her time and effort in reviewing our manuscript entitled "The Effects of Varying Glenohumeral Joint Angle on Acute Volume Load, Muscle Activation, Swelling and Echo-Intensity on the Biceps Brachii in Resistance-Trained Individuals” On this revised version, we attempted to address most of the points and edits brought up by the reviewer. The modifications throughout the manuscript in response to the comments listed below are highlighted in yellow.
Line 20: please format the upper case for °.
Answer: It was changed accordingly.
Line 26-27: state more practical outcome
Answer: We have added some practical applications to our conclusion section.
Line 34-35, 38-40: The exercise tempo is very important parameter for acute response as well.
https://www.ncbi.nlm.nih.gov/pubmed/29922395
Answer: We added tempo to our i.e.
Line 93: why you selected tempo 202?
Answer: We used a 202 tempo as it was best maintained with the metronome when piloting and is fairly common tempo used for those training for hypertrophy. We didn’t want to have a concentric or eccentric dominant rep tempo so we used a 1:1 ratio.
Line 176-185: How did you avoided the electrode positioning differences between different angles?
https://www.ncbi.nlm.nih.gov/pubmed/16102976
Answer: We maintained the same electrode position throughout the session. Once the wireless EMG sensor was applied at the 37.5% landmark, it remained untouched throughout the training session.
Line 187: Please report the effect sizes since you have n = 11.
Answer: We reported p values and confidence intervals for all variables, however we felt the use of effect sizes was only needed to express the magnitude of measurement for echo-intensity.
Line 226: the figure 4 is not reported.
Answer: It has been changed accordingly and all figures are up to date.
Round 2
Reviewer 1 Report
Manuscript ID: sports-576048-v2
Thank you for giving me the opportunity to review this manuscript.
COMMENTS TO THE AUTHOR(S)
I fully agree with the authors on the normalization method they chose for EMG amplitude during the dynamic tasks. However, this is not related to the issue of the IZ. The authors cannot exclude that during the dynamic task the IZ moved under the electrodes. The best position for EMG acquisition, particularly in bipolar mode, would have been more proximal to the cubital fossa. Anyway, I suggest to clearly state this concern in the limitations of the study.
Other comments:
P4 L155 Please add the electrodes size and inter-electrode distance.
Where the signals acquired in bipolar mode?
P5 Figure 2 Please describe what the circle means.
P9 L310 “The electrode placement is still debated in the literature [38]”.
Not clear, please rephrase.
Electrode placement for bipolar acquisitions on the biceps brachii was clearly defined (see Barbero, Merletti, Rainoldi 2012. Atlas of muscle innervation zones). Moreover, since the BB is probably the most studied muscle in sEMG acquisitions, the definition of its optimal electrode site was a priority.
Please modify also the reference.
Author Response
Reviewer Comments
Thank you for giving me the opportunity to review this manuscript.
--- We appreciate your input and critiques as it has given us the opportunity to revise and strengthen our paper. Thanks again.
COMMENTS TO THE AUTHOR(S)
I fully agree with the authors on the normalization method they chose for EMG amplitude during the dynamic tasks. However, this is not related to the issue of the IZ. The authors cannot exclude that during the dynamic task the IZ moved under the electrodes. The best position for EMG acquisition, particularly in bipolar mode, would have been more proximal to the cubital fossa. Anyway, I suggest to clearly state this concern in the limitations of the study.
--- We have reviewed the Atlas of Muscle Innervation Zones textbook and better understand your comments and concerns. Thus, we have edited and added to our limitation section to better reflect these issues.
Interestingly, our mid-belly landmark (37.5%) would translate to 62.5% of our anatomical landmark frame (ALF) compared to 61% & 62% for the short head and long head respectively. However, we used the acromion and olecranon compared to the suggested distal biceps tendon for ALF as suggested by Barbero et al. 2012.
Other comments:
P4 L155 Please add the electrodes size and inter-electrode distance.
Where the signals acquired in bipolar mode?
--- This information has been added to the materials and methods section (2.5) as suggested by the reviewer. Yes, the signals were acquired in bipolar mode
P5 Figure 2 Please describe what the circle means.
--- Sorry for the misunderstanding. We just wanted to highlight our landmark for the reviewers. Thus we utilize the yellow circle. If you zoom in on the image you will see additional marking but we felt like this circle makes that more apparent.
P9 L310 “The electrode placement is still debated in the literature [38]”.
Not clear, please rephrase.
Electrode placement for bipolar acquisitions on the biceps brachii was clearly defined (see Barbero, Merletti, Rainoldi 2012. Atlas of muscle innervation zones). Moreover, since the BB is probably the most studied muscle in sEMG acquisitions, the definition of its optimal electrode site was a priority.
--- Please, see our response above. The reviewer is right.
Please modify also the reference.
--- Thanks again for providing that reference. We have edited the limitations section extensively based on the reviewer's comments and the reference provided. We have removed the previous reference and added the aforementioned reference above.
Reviewer 2 Report
None
Author Response
---Thank you for giving us the opportunity to revise and strengthen our manuscript ----